# A Methodology for Evaluating Multimodal Referring Expression Generation for Embodied Virtual Agents

## ABSTRACT

Robust use of definite descriptions in a situated space often involves recourse to both verbal and non-verbal modalities. For IVAs, virtual agents designed to interact with humans, the ability to both recognize and generate non-verbal and verbal behavior is a critical capability. To assess how well an IVA is able to deploy multimodal behaviors, including language, gesture, and facial expressions, we propose a methodology to evaluate the agent's capacity to generate object references in a situational context, using the domain of multimodal referring expressions as a use case. Our contributions include: 1) developing an embodied platform to collect human referring expressions while communicating with the IVA. 2) comparing human and machine-generated references in terms of evaluable properties using subjective and objective metrics. 3) reporting preliminary results from trials that aimed to check whether the agent can retrieve and disambiguate the object the human referred to, if the human has the ability to correct misunderstanding using language, deictic gesture, or both; and human ease of use while interacting with the agent.

## CCS CONCEPTS

• **Human-centered computing** → **HCI design and evaluation methods**; • **Computing methodologies** → **Natural language generation**.

## KEYWORDS

Embodied agents, non-verbal behaviours, multimodality, referring expression generation

**ACM Reference Format:**
Anonymous Author(s). 2023. A Methodology for Evaluating Multimodal Referring Expression Generation for Embodied Virtual Agents. In *Proceedings of Make sure to enter the correct conference title from your rights confirmation emai (Conference acronym 'XX).* ACM, New York, NY, USA, 10 pages. https://doi.org/XXXXXXX.XXXXXXX

## 1 INTRODUCTION

Recent achievements in generative language modeling, of which OpenAI's ChatGPT is an exemplar, have demonstrated remarkable abilities in producing topically coherent, grammatically correct, and contextually appropriate text. Prior to the generative AI boom, language models such as BERT [10] and GPT-2 [54] achieved state

of the art results on various language processing tasks. It may be tempting, therefore, to believe that language generation for conversational agents (CAs) is a solved problem. However, a common critique of large language models (LLMs) is that they lack *grounding* or *understanding*. Bender and Koller [4] argue that learning only from the textual form does not provide information about the "meaning" connecting utterance to communicative intent.

Humans, meanwhile, communicate in multiple non-verbal modalities, and mix these fluently with verbal modalities. A telling example is the ability of a human to answer a question like "what am I pointing at?" with appropriate situational context, which even a multimodal LLM like GPT-4 cannot. Given the recent developments in language modeling, we can expect the ability to fluently mix and match modalities to be a critical capability in the next generation of CAs. As interactive agents become more sophisticated, and see and interpret both visual and linguistic context concurrently, users will expect them to behave more like humans.

Agent embodiment is one channel to provide information needed to enable CAs to understand language in context. If one modality (e.g., language) is not communicative, another modality (e.g., gesture) can be used to disambiguate or correct the failure. As objects in a shared situated context provide anchors for the construction of common ground between interlocutors [7, 50, 51], a valuable use case to understand multimodal language use in context is **multimodal referring expressions** (MREs) that exploit information about both object characteristics and locations [8]. It is therefore necessary to come up with principled strategies to evaluate mixed-modality referring expression generation systems.

In this paper, we propose a methodology to carefully evaluate generation of multimodal referring expressions by a particular class of CAs, namely embodied interactive virtual agents (IVAs), with the goal of aiding the development of IVAs that interact with humans with symmetrical, bidirectional use of non-verbal and verbal behavior. Our novel contributions are:

- An embodied virtual agent testbed with an IVA who uses gesture and language [26, 40] to elicit MREs from humans;
- Establishing bidirectional and symmetric communication between humans and IVAs using verbal and non-verbal behavior synthesis;
- Evaluation metrics thereof that apply to both humans and IVAs, combining qualitative and quantitative metrics;
- Analysis of preliminary data gathered from interactions with our test agent.

## 2 RELATED WORK

The psycholinguistic literature shows the impact of deictic gesture on the successful communication of intent and reference for both speakers and hearers [17, 41]. Nonetheless, much earlier work in the area of referring expression (RE) generation has focused on linguistic description, such as relative and absolute properties of objects

(e.g., size and color) [16, 61], spatial references [12, 32, 37], and relational episodic descriptions [13]. Where non-verbal information, such as deictic gesture, is considered, much prior work focuses on RE comprehension rather than generation, e.g., [5, 35, 52, 57], and additionally typically lacks features related to agent embodiment [22, 23]. Where generation is addressed [13], it is often separated from comprehension. As such, we seek to build and evaluate models for generating MREs that are fluent and clear, and symmetric and bidirectional in the context they exploit when compared to human-generated REs. Doing so requires developing evaluation metrics that indicate when IVA-generated non-verbal behavior provides a meaningful boost in communicative capability compared to verbal behavior only.

*Datasets.* A number of datasets and corpora exist of human-generated descriptions of target objects in visual scenes, including Bishop [18], Drawer [63], GRE3D3 [64], TUNA [16], RS-VS [37], and recent corpora by Kunze et al. [32] and Doğan et al. [12]. Other RE corpora collected for the purpose of training comprehension models fall into three categories—verbal references only [6, 9, 20, 39, 42, 45, 67], gestures only [56, 58, 59], and embodied multimodal REs including language and gesture [30, 55].

*Metrics.* Correspondence between human corpora and machine generated references can be measured either by automatic metrics or human judgments. Overlap in the properties of human and machine descriptions can been computed according to Dice Coefficient [11], MASI [44], Levenshtein Distance [34], BLEU [43], ROUGE [36], CIDER [62], or METEOR [2]. Alternatively, human judges can evaluate generated REs according to adequacy of reference or naturalness. While adequacy is evaluated by object identification tasks [12, 13, 15, 32], naturalness is evaluated by (1) metrics such as error rate, identification time, and reading time [3, 29] or (2) human ranking of generated references for objects in a set of images or videos [12, 30, 32].

Prior work on embodied agents argues for the role of embodiment in representing the salient content of objects in a scene [49], in contributing to mutual understanding [25], and in evaluating the outputs of interactive systems [31]. Relatedly, Kozierok et al. [21] argue that evaluating multimodal interactions require a combination of quantitative and qualitative criteria, particularly in task-based situations. We therefore present a task-oriented setting designed to require the use of MREs, and a proposal for evaluating how non-verbal strategies complement verbal strategies for situated meaning [53].

In the remainder of this paper, we will discuss the platform we use to collect and generate MREs in a human-agent interaction (Sec. 3), specify the evaluation metrics we propose to use (Sec. 4), present preliminary results of initial data collected according to the proposed evaluation (Sec. 5), and discuss future directions (Sec. 6).

## 3 METHODOLOGY

First, we develop an interactive virtual agent system for an object identification task that interprets human language and simulated gesture inputs, and responds with language and animated gestures. We then proposed metrics to address the fluency and clarity of referring expressions used. Since our goal is to create symmetric,

bidirectional communication between humans and agents, these metrics may apply to either human or agent behaviors, and we compare the use of verbal and non-verbal modalities. We then analyze preliminary data for indications of where human and agent use of different modalities aids communication, for the purposes of assessing the contribution of non-verbal behavior to the interaction.

### 3.1 Interactive Virtual Agent (IVA) Development

The *Diana* system [26, 47] was developed as a collaborative virtual agent who responds to instructions given via both live gesture and speech and collaborates with humans in situated task-based interactions. We adapted the existing system into a standalone version where human participants are presented with a sequence of 10 scenes, each involving (1) ten equally sized target blocks randomly placed on a table that (in simulated units in the Unity-based environment) is approximately 1.6m wide. There are two of each color of block: red, green, blue, pink, and yellow; and (2) two landmark objects (*plate* and *cup*) available for use when describing the target blocks. This setting requires the IVA to ask for disambiguation based on factors like color and location if needed, and the human to provide complex descriptions including verbal (e.g., relational, historical) references, non-verbal (e.g., deictic pointing) references, or ensemble. Diana initially asks a question, e.g., "Which object should we focus on?", as shown in Fig. 1, without providing any prior knowledge of what she understands, e.g., specific domain words or actions. Participants are informed that they are able to use multiple input channels, e.g., automatically recognized speech and mouse-based deixis, to clearly express their intent. To replicate the variability in pointing displayed in the Diana system with live gesture recognition, and the gesture-semantic notion of a *pointing cone* [24], the center of deixis fluctuates within a circle of radius ±0.3m around the mouse location and the size of the deictic reticle (see Fig. 1) randomly fluctuates in size within a range of 14–186% of the default radius (17.32cm). This variability prevents users from relying on fully accurate pointing with the mouse as a method of unambiguously indicating objects, and encorages the use of speech input for object specification.

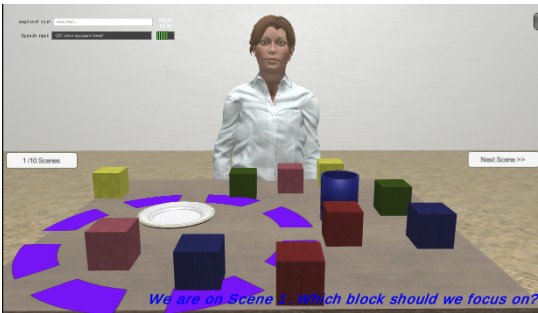

**Figure 1: Experimental Diana System: the purple circle indicates where the user is pointing. Without disambiguation, any object within the pointing circle is a potential candidate for a deixis-only RE. Diana's utterances both appear on screen and are spoken aloud via TTS.**

**Table 1: Predicate logic format (PLF) transformation for co-gestural verbal REs (Att _RE: Attributive RE, Trans_RE: Transitive RE, Rel_RE: Relational RE, Hist_RE: Historical RE, and Comp_RE: Compound RE). *Numerals in brackets denote variables that must be assigned from prior conversational or non-verbal context (e.g., "it," "there," etc.).**

| Speech Prompt | PLF | Verbal | Non-Verbal | RE Type |
|---|---|:---:|:---:|---|
| Pick up that red block | $grasp(that(red(block)))$ | ✓ | ✓ | (Att_RE) |
| Put this block to the right of the blue block | $put(this(block), right(the(blue(block))))$ | ✓ | ✓ | (Trans_RE) |
| Grasp the green block beside the plate | $grasp(the(green(beside\_adj(plate(block)))))$ | ✓ | - | (Rel_RE) |
| Lift the block you just put down | $lift(the(put\_adj(block)))$ | ✓ | - | (Hist_RE) |
| Take this block and put it there | $take(this(block)) + Put(\{0\}, \{1\})^*$ | ✓ | ✓ | (Comp_RE) |

*Interpreting Verbal and Non-Verbal Expressions.* Multimodal referring expressions can be considered special cases of *gesture utterances* as specified in [48], in that they contain a gestural component and a verbal component that must be unified for a complete interpretation by either human or machine. In addition, MREs may be mixed with unimodal REs in a discourse, but even unimodal REs may rely on meaning that was previously established in the discourse using multimodal communication. Therefore, our motivation for developing a bidirectional evaluation scheme is to create methodologies for evaluating combined verbal and non-verbal behavior that apply equally well to human and IVA behaviors.

We follow an analysis of the EGGNOG dataset, a collection of human-human interactions in a Blocks World domain [65], wherein human-generated verbal REs are expected to fall into three complex categories, potentially involving both verbal and non-verbal content: *Attributive REs*, which describe object properties; *Relational REs*, which describe objects in relation to each other; and *Historical REs*, which describe objects already mentioned or interacted with. All three of these may be aligned with deictic gesture, but in different ways. To replicate these exhibited interpretive capabilities, we first developed four main algorithms to interpret verbal REs: (1) *<ParsingToPLF>* recursively follows a set of rules, using the Stanford CoreNLP dependency tree [38] to compose linguistic constituents into a predicate logic format (PLF). Table 1 shows the PLFs of different speech inputs and whether they need to be accompanied by non-verbal information for a complete interpretation. Multimodal references are interpreted with respect to the VoxML modeling language [33, 46] and the scene in the VoxWorld simulation platform [27, 28]. (2) The *<RelationalRE>* algorithm leverages spatial relations between objects that are tracked by the VoxWorld platform using calculi such as RCC-3D [1]. The interpreter extracts mentioned objects, localizes the target relative to other objects, and acts upon it as shown by command #7 in Fig. 2. (3) The *<HistoricalRE>* algorithm processes those sub-predicates that indicate actions that have previously been taken in the dialogue, e.g., in Fig. 2 #9, by extracting objects that were the subjects of previous actions. (4) The *<AttributiveRE>* algorithm interprets attributes of the object (in this setting, color or demonstratives). There are two subcategories of attributive REs: *Transitive REs*, an attributive RE embedded within a transitive clause (see Table 1); and *Compound REs* involving multiple events, as shown by Fig. 2 #1. A simple, non-transitive, non-compound attributive RE is shown in Fig. 2 #3. This utterance also shows how demonstratives may involve the human deictically disambiguating the target. Deictic

inputs can also be sequential pointing prompts with no speech, e.g., Fig. 2 #13–16, in which the human points to *BlueBlock* then to the *GreenBlock* to instruct Diana to put the blue block on the green block.

HUMAN:  [1] Take the red block and put it to the left of the plate.
DIANA:  [2] Which red block?
HUMAN:  [3] This one. [*Points to a red block*]
DIANA:  [4] OK! [*Acts on the red block*]
HUMAN:  [5] Grasp this pink block and put it on the red block. [*points to a pink block*]
DIANA:  [6] Do you mean the red block I just put down? [*Points to a red block*]
HUMAN:  [7] No, the red block behind the blue block.
DIANA:  [8] OK! [*Acts on the red block*]
HUMAN:  [9] Pick up the block that you just grasped.
DIANA:  [10] I've just grasped the pink block (that is) on the red block. Is this what you mean? [*points to a pink block*]
HUMAN:  [11] Yes.
DIANA:  [12] OK! [*Acts on the pink block*]
HUMAN:  [13] [*Points to a blue block*]
DIANA:  [14] [*Points to the same blue block*]
HUMAN:  [15] [*Points over the green block*]
DIANA:  [16] [*Puts the blue block on the green block*]

**Figure 2: Sample dialogue: the interaction from 1–12 is multimodal (co-gestural speech) and from 13–16 is unimodal (deictic gesture only).**

*Generating Verbal and Non-Verbal Expressions.* In addition to interpreting multimodal inputs, being able to generate non-verbal behavior is essential for interactive agents to add social fluency to the interaction [66]. Diana is able to generate speech via text-to-speech, deictic gesture via animation and inverse kinematics executed on her body rig, and action by manipulating virtual objects in the scene. (1) When the human indicates a block without supplying an action to execute, Diana points to it, confirming understanding of the RE with her own deictic RE, as shown in Fig. 3. (2) She directly acts on all aforementioned verbal prompts (e.g., multimodal commands in Fig. 2, #1–12) by either disambiguating candidate target objects or carrying out the requested action in the virtual space. (3) She also acts on non-verbal prompts (e.g., unimodal commands in Fig. 2 from 13-16) by performing the denoted actions after the human specifies the focus and target locations. (4) As shown in Fig. 4, she expresses emotions (e.g., confusion and joy), in response to human inputs, such as being confused when there is an ambiguity in RE or action interpretation, or joy at having interpreted an input successfully. Appropriate generation, then,

becomes a question of correctly generating the content of an utterance, movement through space of a gesture, or specific facial expression at the right time, to serve a communicative purpose.

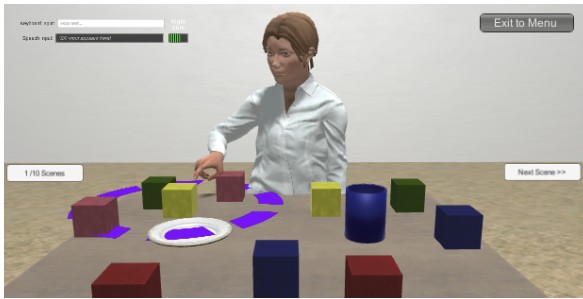

**Figure 3: Generating deictic gestures. Diana will respond to what she interprets the RE as referring to by pointing to it, which can be used to assess the correctness of her object grounding depending on which object the human actually intended to reference.**

## 4 EVALUATION

With the goal to enable bidirectional communication between machines and humans using multimodal referring expressions as a testbed use case, specific evaluable properties must be enumerated to demonstrate where a fully-symmetrical system is more successful than one that maintains communicative asymmetry between the two interlocutors. The key research question with evaluation is: *do the metrics used clearly establish whether both interlocutors are able to extract the communicative intents of the others from their behavior?* Therefore, good metrics will answer if the non-verbal behavior generation methods used for an IVA is effectively contributing to the human interlocutor's understanding, as defined as the ability to extract communicative intent from utterances and actions. We consider properties that are related to deictic and linguistic context awareness, as used in the evaluation of human-machine collaboration [21], and propose quantitative and qualitative metrics that assess the following properties of multimodal RE usage in a task-based environment: 1) efficient and collaborative task completion, 2) software reliability and consistency, 3) ability of humans and machines to understand diverse communications, and 4) agent contribution of meaningful content. The version of the Diana system described above is presented to human subjects to collect samples

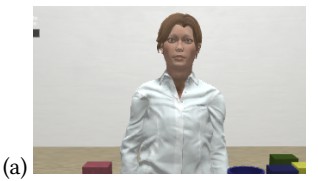
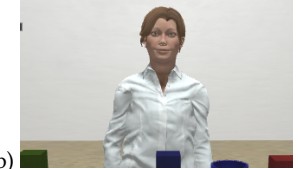

(a)                    (b)

**Figure 4: Diana's facial expressions. (a) Confusion (e.g., undoing an action or responding to a negative acknowledgment). (b) Joy (e.g., welcoming users at the beginning of interactions or responding to a positive acknowledgment).**

of bidirectional collaborations and evaluate successful multimodal communication strategies for RE generation using both logged interactions and human judgments.

### 4.1 Human-Machine Collaboration Data Collection

During a single human-agent interaction session, the participant views 10 scenes containing 10 randomly-placed target objects to be referenced. Referencing is considered successful when Diana is able to ground the human's MRE to the same object as the human intends to describe. The IVA's and participant's utterances, non-verbal behavior, and actions are logged (e.g., Fig. 5) for analysis and future training and evaluating of multimodal referring expression generation models.

### 4.2 Evaluation Metrics

To evaluate the success of the IVA w.r.t. the key characteristics of human-machine collaboration from Sec. 4, we define 19 metrics as follows:

(1) Multimodal Prompt Completion Efficiency (MPCE).
(2) Linguistic Prompt Completion Efficiency (LPCE).

The difference in target identification and the related task completion times when using multimodal REs vs. verbal only REs indicates the increase in RE effectiveness when using multimodal generation vs. linguistic generation methods only.

(3) Human-machine completion efficiency (HMCE): Time taken to complete the task. Since the task as a whole is normalized (an object referencing with 10 scenes each containing 10 objects), completion time can be directly related to referring strategies used by each interlocutor.
(4) Machine Appropriate Response Success Rate (MARSR): Rate of IVA responses to human prompts that are not followed by a negative response (e.g., no, nevermind).
(5) Proceed Without Reset (PWR): Rate of interactions that proceed without resets.
(6) Machine Interpretation of Human Communication (MIHC): Rate of correctly executed prompts.
(7) Machine Interpretation of Relational REs (MIRRE): Rate of correctly executed relational prompts.
(8) Machine Interpretation of Historical REs (MIHRE): Rate of correctly executed historical prompts.
(9) Human Interpretation Efficiency of Machine Communication (HIEMC): Time from generation of machine's reference to target identification by human.
(10) Agent Pointing Success Rate (APSR): Rate of agent successfully pointing out the target object.
(11) Mutual Contribution Success Rate (MCSR): Difference between number of verbose human turns and verbose agent turns ("verbose" being defined as a meaningful contribution beyond positive or negative acknowledgement or disambiguatory question—in our MRE use case this typically means a distinct referring expression).
(12) Machine-generated referring expressions (MGRE): Rate of machine-generated referring expressions compared to total utterances/discourse moves.

(13) Recognition of Previously Mentioned Entities (RPME): Rate of previously mentioned entities grounded at the end of each discourse move.

(14) Machine Historical Referencing Success (MHRS): Rate of historical references generated by the agent relative to total number of generated REs.

(15) Machine Relational Referencing Success (MRRS): Rate of relational references generated by the machine relative to total number of generated REs.

The above metrics 1–15 are all calculated directly from data logged during human-agent interactions. The following metrics are collected *post facto* from the judgments of 3rd-party evaluators (see Sec. 6.1).

(16) Machine Object Identification Success Rate (MOISR): Rate of correctly identified objects (by machine).

(17) Human Object Identification Success Rate (HOISR): Rate of correctly identified objects (by humans).

(18) Machine References Fluency Rate (MRFR): Rate of top-rated machine references according to 3rd-party human judgments.

(19) Human References Fluency Rate (HRFR): Rate of the top-rated human references according to 3rd-party human judgments.

In this paper, we include preliminary results for the following metrics: Multimodal Prompt Completion Efficiency (MPCE), Human Interpretation Efficiency of Machine Communication (HIEMC), and Agent Pointing Success Rate (APSR), in addition to the illustrations of generated referring expressions by each of the IVA and subject, IVA's ability to disambiguate, human's ability to correct IVA's misunderstanding, the impact of deictic gesture on interlocutors' understanding, and IVA's dialogue history.

## 5 PRELIMINARY RESULTS

### 5.1 Automated Quantitative Evaluation

In a preliminary study, constituting the complete 10-scene interaction with a sample test subject, we logged 330 different human referring expressions, including 141 pointing-only references for target object identification, 141 pointing-only references for target location identification, 33 multimodal REs, and 15 linguistic REs, as depicted in Fig. 6a. Linguistically, as shown in Fig. 6c, 84% REs are transitive attributive references (e.g., *move the red block to the plate*). Similarly, we logged 330 different machine referring expressions, including 141 pointing-only REs to the referents, 174 multimodal REs, and 15 linguistic REs, as depicted in Fig. 6b. Consequently, we used these logged data to obtain preliminary results regarding the ease of agent disambiguation, human recognition of agent intent from verbal and non-verbal behavior, and overall interaction.

In Fig. 5a, interlocutors' moves, including actions, speech, and gestures, are logged with their timestamps. We see that the human started pointing to the focus object (*BlueBlock1*) and moving it behind *YellowBlock1*. Logs also include the positions of each, distance from agent to each, and the agent's action after pointing to each of the two blocks. The human then used language only ("Pick up the yellow block") to instruct Diana to pick up *YellowBlock2*. This instruction required Diana ask for disambiguation: "Which yellow block?", as there are two yellow blocks in the scene. To disambiguate, the human uses pointing, and the object, its position,

(a)

```
[2023-06-07-11-51-05] ————————————————————Pointing to the FOCUS without Speech—————
[2023-06-07-11-51-05] user:intent: object _ focus  |  BlueBlock1
[2023-06-07-11-51-05] Focus object position  |  (0.2, 1.1, -0.3)
[2023-06-07-11-51-05] Distance from agent to focus obj:1.188692
[2023-06-07-11-51-05] Diana pointed | BlueBlock1
[2023-06-07-11-51-06] ————————————————————Pointing to the TARGET without Speech ————
[2023-06-07-11-51-06] target object  |  YellowBlock1
[2023-06-07-11-51-06] target object position  |  (0.0, 1.1, -0.2)
[2023-06-07-11-51-06] Distance from agent to focus obj:1.188692
[2023-06-07-11-51-06] user:intent:event  |  put(BlueBlock1,behind(YellowBlock1))
[2023-06-07-11-51-06] agent executed | put(BlueBlock1,behind(YellowBlock1))
[2023-06-07-11-51-06] —————————————————————————————————————————
[2023-06-07-11-51-06] Relations: under | Table and PinkBlock1 + under | Table and BlueBlock2 ...
[2023-06-07-11-51-06] Configurations: RedBlock1: (0.583337500: 1.124870000: 0.429246700) ...
[2023-06-07-11-51-06]—————————————————————————————————————————
[2023-06-07-11-51-07] user speech  |  pick up the Yellow Block
[2023-06-07-11-51-07] Parsed speech  |  grasp(the(yellow(block)))
[2023-06-07-11-51-07] Diana  |  Which Yellow Block?
[2023-06-07-11-51-08]—————————————————————Pointing to the FOCUS After Disambiguation——
[2023-06-07-11-51-08] user:intent: object _ focus  |  YellowBlock2
[2023-06-07-11-51-08] Focus object position  |  (0.5, 1.1, 0.1)
[2023-06-07-11-51-08] Distance from agent to focus obj:1.447748
[2023-06-07-11-51-09] Diana executes | grasp(the(yellow(block)))
```

(b)

```
[2023-06-07-13-38-16] User speech  |  Grasp this block
[2023-06-07-13-38-16] Parsed speech  |  Grasp(this(block))
[2023-06-07-13-38-17] Diana  |  OK!
[2023-06-07-13-38-17] User speech  |  No, this block
[2023-06-07-13-38-18] Diana  |  OK!
[2023-06-07-13-38-39] User speech  | Move the green block to the left of the plate
[2023-06-07-13-38-39] Parsed speech  |  slide(the(green(block),left(the(plate)))
[2023-06-07-13-38-39] User speech  |  nevermind
[2023-06-07-13-38-40] Diana  |  OK! Nevermind.
```

**Figure 5: (a) Trial sample of Diana's ability to disambiguate the target; (b) Trial sample of human's ability to correct misunderstanding.**

and distance are logged, along with Diana's action. This illustrates Diana's capability to clearly disambiguate the object the human referenced and efficiently execute the human's prompt as shown in Fig. 7a and b, which leads to bidirectional communicative efficiency, with both human and agent combining verbal and non-verbal behavior. When Diana has a misunderstanding, the human can correct it using language, deictic gesture, or both (Fig. 5b). Diana confirms that disambiguation was successful using deictic gesture to the correct object.

In human-human interactions, pointing reduces cognitive load [17]. Similarly, this is observed with the IVA as shown in the contingency table, Table 2. The agent shows her understanding of the human's intended meaning when providing a sequence of pointing REs or co-gestural speech (Multimodal REs) without asking for disambiguation by pointing to the referents; nonetheless, using only speech for communication requires the agent to ask for additional information, i.e., gestures, to clearly identify the target and point to it as depicted in Fig. 7c. We see that a relationship exists between the modalities used and the level of ambiguity, such that use of pointing significantly reduces the ambiguity level of the prompt ($p$-value $< 0.001$ using Fisher's exact test [14]).

In addition to language and deictic gesture, prior actions contribute to building speakers' knowledge of descriptions of objects as defined by Grice's maxim of quantity [19]. Therefore, we integrated a dialogue history to the IVA. This stack stores all requested actions along with target objects, and accomodates interpretations of verbal, gestural, and multimodal inputs. Fig. 8 shows the number of actions in the dialogue history by the end of each scene in the preliminary data. These stored actions are available for use by both

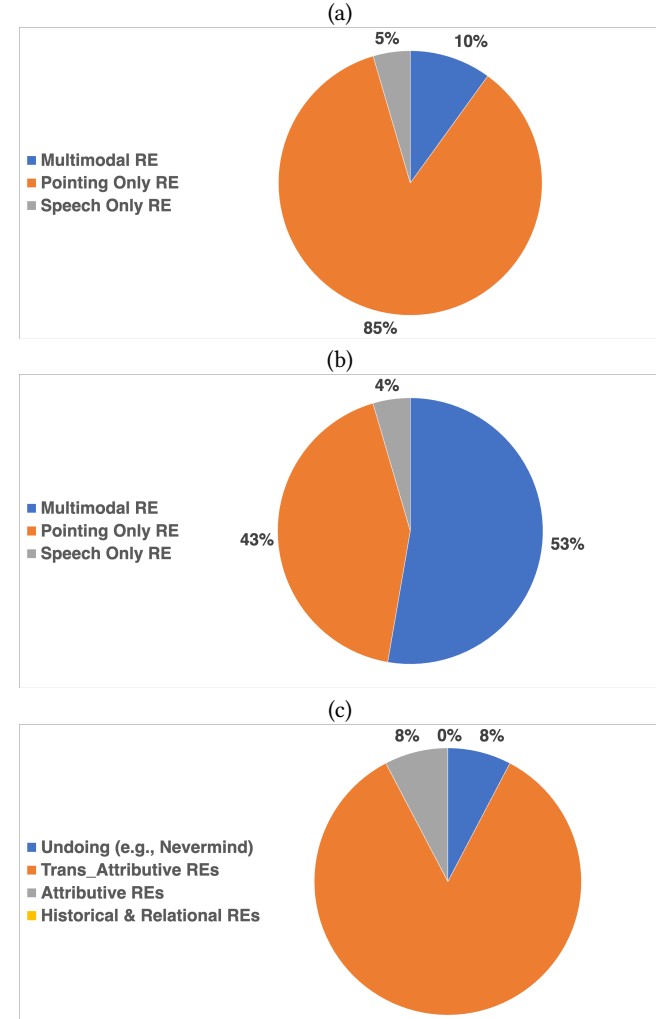

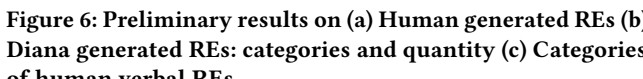

Figure 6: Preliminary results on (a) Human generated REs (b) Diana generated REs: categories and quantity (c) Categories of human verbal REs.

Table 2: Contingency table of human RE ambiguity and modalities used: # ambiguous REs by modality type

| Modality | Did Agent Disambiguate? | |
|---|---|---|
| | No | Yes |
| Multimodal RE | 15 | 0 |
| Pointing Only RE | 141 | 0 |
| Speech Only RE | 0 | 33 |
| $p$-value | $< 2.2e-16$ | |

humans and the IVA to refer to objects that may have previously been interacted with, as described in Sec. 3.1.

Table 3 shows how the IVA's dialogue history is constructed and revisited to understand the human's intents within a shared space.

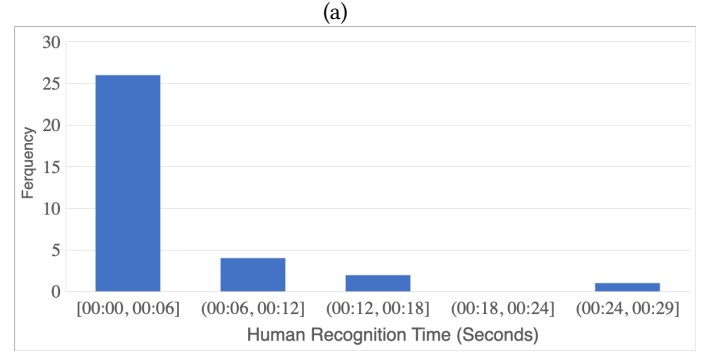

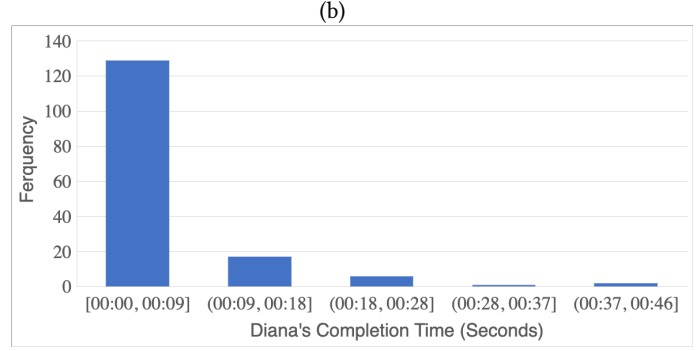

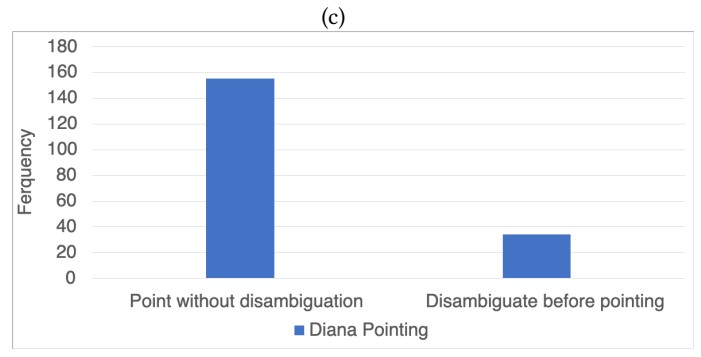

Figure 7: (a) Human Interpretation Efficiency of Machine Communication (Metric #3: HIEMC); (b) Multimodal Prompt Completion Efficiency (Metric #1: MPCE) by Diana; (c) Agent Pointing Success Rate (Metric #10: APSR).

After recognizing the human's intent and executing the parsed-out prompt, the IVA pushes the action and referent (extracted from the PLF of the prompt) to two separate stacks (an actions stack and a referents stack) as shown by Table 3, #1–3. If the human uses a mention of a previously executed action to indicate an object as in Table 3, #4 ("grasp the block you just slid"), the IVA visits the dialogue history to 1) retrieve the most recently referenced object that is relevant to the provided action (in this case, *GreenBlock2*, as it satisfies the *adj_slid*($\cdot$) predicate), 2) push the new most recent action and referent onto the stack for future retrieval if necessary.

**Table 3: Sample of dialogue history, including previously mentioned actions and related objects after executing multimodal (co-gesure speech) or unimodal (speech only or pointing only) prompts.**

| No. | Modality | PLF | Actions Stack | Referents Stack |
|-----|----------|-----|---------------|-----------------|
| 4 | Speech Only | $grasp(the(adj\_slid((block))))$ | grasp put put | GreenBlock2 RedBlock1 GreenBlock1 |
| 3 | Multimodal | $slide(GreenBlock2; left(the(plate)))$ | slide put put | GreenBlock2 RedBlock1 GreenBlock1 |
| 2 | Pointing Only | $put(RedBlock1; left(the(plate)))$ | put put | RedBlock1 GreenBlock1 |
| 1 | Pointing Only | $put(GreenBlock1; < 0.5919505; 1.12487; -0.3801433 >)$ | put | GreenBlock1 |

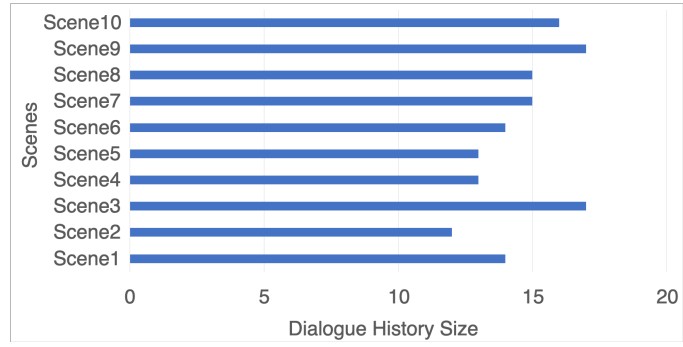

**Figure 8: IVA's dialogue history length at end of each scene.**

## 6 FUTURE EVALUATION

A larger study is preparation with a goal to collect data from roughly 150 participants who use REs of different types and strategies while collaborating with Diana to perform the task described above. Each participant views 10 scenes to refer to 10 randomly placed target objects, resulting in a total of 15,000 samples and recorded videos. Recorded video will consist of screen captures showing the human instructions as they are rendered in the scene, but direct video of the participants will not be collected. The gathered data will then be used to train generative models (e.g., fine-tuning an open-source large language model such as LLaMA [60] or similar) to produce contextually correct and situationally fluent REs that combine language and gesture. These REs will be evaluated according to the metrics discussed above, as well as human judgments as described below.

### 6.1 Human Evaluation

To evaluate the success of multimodal referring expression generation (MREG) models, two human-based experiments will be conducted using crowdsourcing platforms such as Amazon Mechanical Turk (AMT). We propose two primary criteria to assess how generative modules imbued with situational awareness and the ability to prompt non-verbal behavior could be compared with humans' generation capabilities. Criterion 1: how well the agent-generated strategies *qualitatively* compared to humans-generated strategies, as evaluated using a preference ordering method; Criterion 2: how well the agent-generated multimodal references *quantitatively* compared to humans-generated multimodal references, as evaluated using task completion. Fig. 9 shows the MREG evaluation framework including the design, participants and procedures.

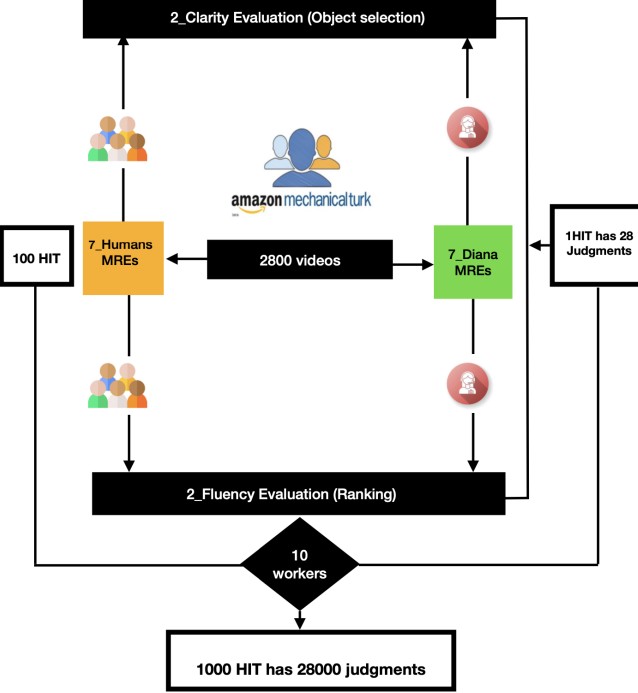

**Figure 9: Crowdsourcing framework for evaluating multimodal referring expression generation models.**

### 6.2 Study Design

Human MREs will be selected from the data gathered according to the strategy outlined in Sec. 4.1. These will be compared with REs generated by the virtual agent when driven by a generative model trained over the human data. A total of 2,800 videos (7 references × 10 blocks × 20 configurations × 2 agents—human and Diana) will be collected. The 7 referencing strategies for each target object will use pointing only once, speech only three times, and a multimodal ensemble three times. This follows the pattern established for data collection in Krishnaswamy and Putejovsky's EMRE dataset [30] which allows for variability in the language used in linguistic or multimodal REs. Videos will be used in a set of AMT human intelligence tasks (HITs), where each HIT will involve workers rating 28 videos for *both* fluency and clarity, including 7 machine generated REs and 7 human REs, for a total of 100 HITs. Each HIT will be completed by 10 workers, for a total of 1,000 HITs and 28,000 individual judgments (2,000 for each individual RE in the dataset). Recruited

**Figure 10: Each set in the HIT includes two tasks for quantitative and qualitative evaluation of human REs and IVA REs.**

workers will be fluent English speakers between 18 and 60 years old and be given 15 minutes for each task while being compensated for their time via the platform.

Each HIT will require workers to evaluate 2 sets of 14 videos according to both the aforementioned criteria (Sec. 6.1). Each set will contain 7 videos of human REs and 7 of machine-generated REs. Workers will be informed whether the descriptions are generated by humans or by the embodied agent. As shown in Fig. 10, first participants will be asked to rate the "fluency" of each description in the video using a Likert-type scale (from 5—best—to 1—worst). Then they will be asked to locate the target object that is mentioned by the video, which will be compared to the actual object that was intended to be referenced, as stored in the dataset. This assesses the correctness of the referring expression: does a human listener correctly retrieve the object that was intended to be referenced, and how do verbal and non-verbal signals each contribute to the ability to correctly retrieve the object from the referring expression provided?

## 7 CONCLUSION

As interactive agents become more widespread in everyday use, developers will need principled ways of evaluating their behavior. Modern generative large language models already demand new methods of evaluation beyond metrics such as accuracy, precision, and recall on benchmark datasets. Factors such as fluency, reliability, correctability, and ease of use must be taken into account. This is doubly the case when non-linguistic modalities are involved, as would be the case with *embodied* IVAs. In this paper, we proposed a quantitative and qualitative evaluation framework to assess the quality of generated multimodal referring expressions, including language, gesture, and actions grounded in a shared virtual environment. We developed an instance of an IVA for an object referencing task designed to elicit multimodal referring expressions from human interlocutors and developed a set of metrics for evaluating the quality of referring expressions that apply equally to those produced by both humans and humanoid IVAs using combined verbal and non-verbal information. We showed preliminary results from naive users of the experimental platform, and analyzed system outputs based on a subset of our proposed metrics to showcase their utility for evaluating the contribution of non-verbal information toward bidirectional interpretation and disambiguation of definite descriptions of objects in context. We also detailed how our preliminary study will be expanded and scaled up. Our framework targets both timing and fluency of the interaction and proposes a set of qualitative and quantitative metrics that we hope will be beneficial for researchers in the IVA and multimodal interaction communities to assess dialogue and behavior generation strategies for multimodal interaction systems.

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

Received 21 July 2023

