# OpenReview forum: "A Methodology for Evaluating Multimodal Referring Expression Generation for Embodied Virtual Agents"
_ACM.org/ICMI/2023/Workshop/GENEA — GENEA Workshop 2023_

### Official Review · Reviewer_6dHr · 2023-08-08
**Missing details lead the novelty to be hardly evaluated**

**Rating:** 4
**Confidence:** 2

**Review:**

This paper intends to propose an evaluation method for the interaction between virtual agents and humans. The author presented an interactive process that asks or indicates the virtual agent for the specific response while the virtual agent will respond based on the question and the world.  Later, they propose 19 evaluation metrics, but only 4 are applied in this paper. Personally, the research problem is interesting. Finally, they apply user study to evaluate However, the missing details lead the paper's contribution and novelty to be hardly evaluated.

Clarity: The paper is not easy to follow.
- Some abbreviations are not well defined when they appear the first time.
- Missing the description of the MPCE and  LPCE.
- Figure 6 and 7 is confusing: It is not clear why fig. 6(c) is important to the task?, it is also confused that why fig. 7 (a) and (b) use a different recognition time.
- Why show the human recognition time and Diana's completion time in fig 7 (a) and (b)?
- Why 19 metrics are proposed but only applied 4 in the paper?
- What is the connection between user study and those proposed evaluation metrics?

pros:
- The problem is interesting. getting human to involve for evaluating the generated deictic gesture is interesting.
- The crowdsourcing framework is clear.

cons:
- The evaluation method part is confused.
- It is unclear why 19 metrics are proposed but only 4 metrics are applied.
- Missing details lead the paper hard to follow.

---

### Official Review · Reviewer_azKH · 2023-08-08
**The paper developed an embodied platform to collect human robot interaction for a specific domain, multimodal human referring expressions. It also offers several qualitative and quantitative evaluation metrics based on the developed framework.**

**Rating:** 7
**Confidence:** 4

**Review:**

Generally, the paper is well-written and structured with some merits. The proposed metrics for evaluating the fluency and clarity of REs are well-defined.



The paper needs a more detailed discussion of the limitations of the proposed methodology, I.e. is it possible to generalize the results to other tasks and contexts? The paper proposed 19 evaluation metrics while reporting preliminary results on a small subset and while it could benefit from more. More explanation of the crowdsourcing platform used to collect human-generated RE, including ethics or any potential biases or limitations of the platform. Some sentences need justification or support from the literature, e.g. ‘’’...  LLM like GPT-4 cannot” or “... LLMs lack understanding” needs a reference.

---

### Decision · Program_Chairs · 2023-08-11

**Decision:**

Accept

**Comment:**

In this paper a methodology for evaluating multimodal referring expression generation for embodied virtual agents is described. The paper describes the use of a virtual agent testbed, with an agent that uses gestures and language to elicit multimodal referring expressions, establishing bidirectional communication, proposes new evaluation metrics and a preliminary study evaluating a subset of these metrics.

One reviewer agrees on accepting this paper, whereas the other is against accepting this paper. Both reviewers indicate a lack of details, whereas one reviewer views this as problematic. Since this is a workshop contribution, I do believe that this work can be worth discussing during the workshop, even when not all metrics are evaluated (which is not something we could reasonably expect from a workshop contribution).